# Fusing Feature Distribution Entropy with R-MAC Features in Image Retrieval

**DOI:** 10.3390/e21111037

**Published:** 2019-10-25

**Authors:** Pingping Liu, Guixia Gou, Huili Guo, Danyang Zhang, Hongwei Zhao, Qiuzhan Zhou

**Affiliations:** 1College of Computer Science and Technology, Jilin University, Changchun 130012, China; gougx18@mails.jlu.edu.cn (G.G.); huili6@staff.weibo.com (H.G.); zhangdy19@mails.jlu.edu.cn (D.Z.); zhaohw@jlu.edu.cn (H.Z.); 2Key Laboratory of Symbolic Computation and Knowledge Engineering of Ministry of Education, Jilin University, Changchun 130012, China; 3School of Mechanical Science and Engineering, Jilin University, Changchun 130025, China; 4College of Communication Engineering, Jilin University, Changchun 130012, China; tongxin@jlu.edu.cn

**Keywords:** image retrieval, pooling method, convolutional neural network, feature distribution entropy

## Abstract

Image retrieval based on a convolutional neural network (CNN) has attracted great attention among researchers because of the high performance. The pooling method has become a research hotpot in the task of image retrieval in recent years. In this paper, we propose the feature distribution entropy (FDE) to measure the difference of regional distribution information in the feature maps from CNNs. We propose a novel pooling method, which fuses our proposed FDE with region maximum activations of convolutions (R-MAC) features to improve the performance of image retrieval, as it takes the advantage of regional distribution information in the feature maps. Compared with the descriptors computed by R-MAC pooling, our proposed method considers not only the most significant feature values of each region in feature map, but also the distribution difference in different regions. We utilize the histogram of feature values to calculate regional distribution entropy and concatenate the regional distribution entropy into FDE, which is further normalized and fused with R-MAC feature vectors by weighted summation to generate the final feature descriptors. We have conducted experiments on public datasets and the results demonstrate that our proposed method could produce better retrieval performances than existing state-of-the-art algorithms. Further, higher performance could be achieved by performing these post-processing on the improved feature descriptors.

## 1. Introduction

Content-based image retrieval (CBIR) has achieved appreciable performance over its long-standing development and has attracted more and more attention among researchers in recent years [1,2,3]. It aims to search the images with the same object, instance, and architecture from an image database and rank the images from the database to certain query images according to the similarities. The global features extracted from visual clues like texture and color were utilized to realize image retrieval in early times [1,2,4]. However, the global descriptors might change with the illumination, occlusion, and translation, and it is hard to keep invariance, which would reduce robustness and affect the performance of image retrieval. Later, the local descriptor of scale-invariant feature transform (SIFT) was proposed to meet invariance exception [5]. The appearance of SIFT has spawned a heavy load of excellent algorithms, which have achieved effective performance in image retrieval. At the beginning, most of the SIFT-based image retrieval methods relied on the bag-of-visual-words (BoW) model to obtain a compact vector of images [6]. Later, the vector of locally aggregated descriptors (VLAD) was proposed to consider all of cluster centers and the distance of local features to its nearest cluster center [7]. Fisher vector (FV) calculates the distances of local features to all cluster centers [8]. Then, spatial pyramid matching (SPM) was proposed, which is based on BoW, with spatial location information added to the feature descriptors [9]. All these methods fail to extract high-level semantic features.

Recently, convolutional neural network (CNN) has made a huge development with the success of AlexNet in the task of image classification [10], and has been widely applied in the tasks of image retrieval [11,12,13,14,15], object recognition [9,16,17], and target detection [18,19]. There are a number of CNN-based methods that have achieved acceptable performance in these tasks, as the deep features extracted from fully connected or convolutional layer contain richer high-level semantic information compared with traditional manual features. Especially for the task of image retrieval, a mass of methods based on the pre-trained network have achieved effective performance [3,12,17,20,21,22,23,24]. Recently, researchers have tended to use the feature maps generated from the last convolutional layer to achieve higher performance in image retrieval [3,12,16,20,21,22,23,24,25,26]. However, the high dimension of feature maps makes the descriptors hard to use directly. In early times, traditional aggregating methods were used to encode the feature map’s output from the CNNs into deep feature descriptors [12,16,26]. With further development of deep learning, more and more methods have been proposed to utilize the feature maps from the CNNs to generate compact feature vectors by using pooling operation [12,20,21,22,23,24,25]. The key challenge for the pooling operation is how to extract the most pivotal features from feature maps and eliminate the effect of the irrelevant information noise. The region maximum activations of convolutions (R-MAC) pooling aim to consider the most prominent points for multiscale regions and has achieved outperforming results in image retrieval [24]. However, this method ignores the difference of distribution in different regions, which could be important to extract more effective feature descriptors for the task of image retrieval. In general, there are some effective post-process methods, like re-ranking [24,25,27] and query expansion (QE) [25,28,29], used in the model image retrieval. These operations could be significantly helpful to further increase the performance of image retrieval.

To solve the key challenge mentioned above, we tend to take the distribution information of feature maps into consideration to generate deep feature representations with richer information. In this paper, we propose a novel method to measure the distribution differences of multiple regions in feature maps called feature distribution entropy (FDE). We combine the proposed FDE with R-MAC to generate more effective features to improve the effectiveness of our image retrieval. To be specific, we make four contributions, as follows.

Firstly, we propose an effective scheme to compute FDE, which could be used to fully reflect the distribution differences of different regions. It would be helpful to focus on the more noteworthy regions and weaken the influence of irrelevant noise.

Secondly, we employ a superior strategy to combine our proposed FDE with R-MAC features to generate more discriminative features. The fused features could tend to extract compact feature representations with more information and are significant to eliminate influence of irrelevant information, especially the noise of background. The compact feature representations are more distinctive and could be more effective in improving the performance of image retrieval.

Thirdly, we perform the operation of re-ranking and QE on the deep-fused features produced by our proposed method. This helps us to obtain better retrieval results.

Fourth, we utilize the fine-tuned network [25] to perform our experiments on different datasets to verify the effectiveness of our proposed method.

To verify the superiority of our proposed method, we perform the experiments on the benchmarks with state-of-the-art re-ranking and QE approaches with the pre-trained and the fine-tuned network. The results of our experiments, which are described in detail in Section 4.3, show that our proposed method outperforms the existing state-of-the-art methods.

We organize the rest of our paper as follow. Section 2 is to illustrate the related work. The calculation of our proposed FDE and the fused features are represented in Section 3. We represent the results and analysis of our experiments in Section 4. Lastly, we make a conclusion for our paper in Section 5.

## 2. Related Work

Deep learning methods based on CNNs have made a great breakthrough in many tasks of computer vision. A general architecture of CNN usually consists of several convolutional layers followed by fully connected layers. The network is usually trained with a softmax layer. Recent works prefer to utilize the activations from the intermediate layer to realize some special tasks like target detection, semantic segmentation, target recognition, and so on, and have obtained effective performance [9,16,17,18,19,30,31]. Particularly for the task of image retrieval, Babenko et al. proposed to use the global features from the fully connected layers in image retrieval [22]. Gong et al. proposed to employ VLAD to aggregate the feature descriptors from the fully connected layers [20]. Recently, more and more works tend to use the feature representations generated by applying pooling on each channel feature maps output from the convolutional layers [3,12,16,20,21,22,23,24,25,26]. These feature representations usually contain richer high-level semantic information than fully connected ones and are significant to promote the effectiveness of image retrieval. More and more works show that better performance of image retrieval could be obtained when the deep feature descriptors are whitened [24,25]. Also, abundant works have shown that some post-process methods, like re-ranking and QE, would be significant to improve the performance of image retrieval [24,25,27,28,29]. In the rest of this section, we describe the related work for the methods we utilize in this paper in detail, which contains the pooling method, normalization, PCA, re-ranking, and QE.

### 2.1. Pooling Approaches

The methods based on CNNs have achieved superior performance in image retrieval. The early works using global features output from the fully connected layers are replaced by the local feature representations derived from the convolutional layers as it has more discriminative descriptive power. In early times, there were some popular encoding methods used in generating compact representations. Gong et al. proposed multi-orderless pooling CNN (MOP-CNN), which aim to extract multi-scale feature maps and utilize VLAD to encode them into the final feature descriptors [20]. Arandjelovic et al. later proposed to apply the VLAD to aggregate local features and design an end-to-end network for image retrieval [16]. Then, Mohedano et al. proposed a novel method, which applies the BOW into deep features [26]. It aggregates the features from CNN into compact representations. Multi-scale feature representation (MFC) was proposed by Hao et al. to extract features from three different scales and fuse the extracted features to generate the final feature vector [32]. However, all these methods mainly use traditional aggregation methods to encode the features from CNN into compact feature descriptors, which are always accompanied by huge consumption of computing. 

There is another way to generate compact representations, which is derived from the pooling layer in CNN. The main idea is to utilize pooling on the activations of convolutional layers to produce more compact deep features [12,20,21,22,23,24,25]. The dimensionality of the deep features is consistent with the channels of feature maps from the corresponding convolutional layer. Babenko et al. propose sum pooling (SPoC), which computes the sum of values in the feature maps [22]. It has shown effective performance in image retrieval. The SPoC feature is calculated as following equation:(1)$$f^{\left(sum\right)}={\left[f_{1}^{sum},\ldots ,f_{c}^{\left(sum\right)},\ldots ,f_{C}^{\left(sum\right)}\right]}^{T},f_{c}^{\left(sum\right)}=\frac{1}{\left|X_{c}\right|}\sum_{x\in X_{c}}x$$
where C is the number of feature maps, c means the channel of features, $f_{c}^{\left(sum\right)}$ denotes the SPoC feature of c-th channel, $\left|X_{c}\right|$ is the amount of feature values in c-th channel feature map, $X_{c}$ is c-th feature map, and x is the feature value in a certain feature map.

Later, Razavian et al. proposed max pooling (MAC) to select the maximum of each feature map [12]. The MAC pooling feature is computed as follows:(2)f(max)=[f1(max),…,fc(max),…,fC(max)]T,fc(max)=xx∈Xcmax
where C is the number of feature maps, c means the channel of feature, $f_{c}^{\left(max\right)}$ denotes the MAC feature of c-th channel, $X_{c}$ is c-th feature map, and x is the feature values in a certain feature map.

According to the former work of MAC pooling, Giorgos et al. proposed R-MAC pooling, which uses sliding windows strategy to obtain a set of regions with different scale [24]. Each region performs MAC pooling in order to obtain the regional feature vectors. The computing equation is shown as follows:(3)fR(r−max)=[fR,1(r−max),…,fR,c(r−max),…,fR,C(r−max)]T,fR,c(r−max)=xcx∈Rmax(r)
where $f_{R}^{\left(r-max\right)}$ denotes the maximum value of the given region, R denotes the regions extracted from the c-th feature map, and $X_{c}\left(r\right)$ means the feature value in the region r on c-th feature map

These methods have made great progress in improving the performance of image retrieval. However, the algorithms described above fail to take the regional distribution information of the feature maps into account. In order to make full use of distribution information, we introduce the concept of entropy to measure the distribution of feature maps. As the R-MAC pooling has obtained superior effectiveness and utilizes multi-scale strategy to extract regions, it is easy for us to analyze their difference. We design an effective scheme to calculate FDE, and then we fuse FDE with R-MAC features. Our experimental results show that our algorithm is better than many existing state-of-the-art algorithms.

### 2.2. Compact Features with Distribution Information

For image retrieval, many traditional algorithms ignore the distribution information of feature maps during generating the deep descriptors. To solve this issue, spatial distribution information is introduced as a supplement to feature descriptors to improve the retrieval performance. Based on BoW, Mehmood et al. combined histograms of local features with global features and constructed local feature maps in local regions [33]. Krapac et al. used Fisher kernel to calculate the spatial mean and cluster changes. Then, they encoded the BoW into a spatial map and combined them with Gaussian mixture model [34]. Koniusze et al. used spatial coordinate coding to simplify spatial pyramid representation [35]. Sancheset et al. improved the performance of FV-based object classification and prompted the spatial position of descriptors [36]. Liu et al. introduced the concept of spatial distribution entropy and applied spatial distribution entropy to the original VLAD algorithm [37]. These methods have achieved great performance in image retrieval, but these improvements were merely applied on traditional algorithms, which are no longer superior to the popular CNN-based algorithms.

Due to the rapid development of neural networks, CNN-based methods have shown excellent retrieval performance in mage retrieval. However, many retrieval algorithms do not make full use of regional distribution information. It is very important to preserve the regional distribution information of images to promote retrieval performance. In R-MAC, the local features of each region are directly concatenated to obtain global features. The contribution of each region is simply the biggest value in each region, which does not consider the difference of regional distribution information and fail to generate more informative feature descriptors. Entropy is an effective measurement to reflect the distribution information of regions, which is proposed by Shannon in 1948 [38]. We take the advantage of the regional distribution information to compute FDE in multiple-scale regions and fuse with the deep feature descriptors as supplement information to solve the disadvantages of R-MAC feature representations. We use FDE to measure the regional distribution information of feature maps and use it as a supplement to combine with the R-MAC features to enrich the deep feature representations. Compared with R-MAC, the results the performance of our algorithm are improved, which indicates that the proposed algorithm is effective.

### 2.3. Normalization and PCA

Normalization plays an important role in image retrieval and has been largely used in image retrieval [13]. This operation aims to transform the data into a uniform scope to make a comparison among them. We would like to discuss two types of normalization, one being L2 normalization and the other being power normalization.

L2 normalization [13] aims to balance the impact of different values, as the values output from the convolutional layer are usually discrete and very different from each other. There would be a mass of extreme values, which would affect the performance of image retrieval. We utilize L2 normalization to narrow the difference of values without changing the proportional difference of values. Specifically, L2 normalization is to limit the values within the range from 0 to 1, and the formulation is defined as follows:(4)$$X_{L2}=\frac{X}{\left\|X\right\|}$$
where ‖X‖ denotes the values in a certain vector and X means the magnitude of this vector.

Power normalization [13] functions the same as L2 normalization to eliminate the gap among extreme values. Power normalization is the reduction of the values in the vector in the form of power exponent, and we give the formulation as follows:(5)$$X_{p}=\mathrm{sgn}\left(X\right)\times X^{p}$$
where X is the values of a certain vector and sig(X) denotes a symbolic function to prevent the sign of the value from changing after power normalization; the value would be 1 if X is larger than 0 and −1 if smaller than 0. p is a hyper-parameter.

The feature vectors generated from the pooling layer tend to have higher dimensions, which could cause large calculation consuming. The features would be accompanied by large noise, which always reduces the performance of image retrieval. To achieve better performance, whitening the feature descriptors is a common and essential stage used in image retrieval as described in the work of Chum et al. [39]. They focus on jointly down-weighting co-occurrences and aim to handle the problem of over-counting. Their work is further migrated in feature descriptors based on CNN. The principal component analysis (PCA) trained on an independent set is always used for whitening and dimensionality reduction [25,40,41]. It aims to project the original vector onto the direction in which the most original information can be retained. The values after PCA are expected to be as scattered as possible with high variance. Mokolajczyk et al. [41] used the training data to whiten local feature representations. Gordo et al. preferred to learn the whitening in an end-to-end manner based on CNN [40], and Filip et al. proposed a new method named learning whitening by taking advantage of training data provided by their 3D models and using liner discriminant projections to perform whitening on features [25]. In our paper, we prefer to utilize PCA to realize dimensionality reduction. It could reduce the computing consuming and eliminate the mutual influence between the original data components to promote the performance of image retrieval.

### 2.4. Re-ranking and Query Expansion

In image retrieval, the results of the first search are often not expressive enough, so reordering the first output will give better results. In image retrieval, re-ranking [27] is often followed by QE [28,29]. The operations of re-ranking and QE are generally helpful in achieving a better performance compared with the results retrieved by raw representations.

Giorgos et al. use approximate max-pooling localization (AML) to coarsely locate the local features of top N images by using the raw representations and then re-rank them [24]. The following QE operation further improves the retrieval performance. In recent years, some new QE methods have been proposed. The most widely used is average query expansion (AQE), which extracts the features of the images ranking top K and averages them with the features of the query images, and then re-retrieving images to obtain a more accurate result. Inspired by AQE, Filip et al. added a weight to the features of the i-th image and named it ɑQE [25].

## 3. Proposed Method

In this section, we give some details of our proposed FDE and introduce how to combine FDE with R-MAC feature vectors to produce more discriminative feature representations, which is significant to improve the effectiveness of image retrieval. Furthermore, we perform re-ranking and QE, which have become standard post-processing used in improving the performance of image retrieval on our proposed method to obtain better performance. We would like to illustrate the process of our method in Section 3.3.

### 3.1. The Algorithm Background

For an image I, we use the pre-trained network without the fully connected layers to output the activations with 3D tensor of W×H×C dimensions, where C means the number of feature maps output from the last convolution layer. For each feature map with size of W×H, which could be represented as $X_{c}$ and k∈{1,…,C}, herein c denotes the feature channel, and each channel feature map with a certain region r, is represented as $X_{c\left(r\right)};$ we denote region location of each feature map as r. To ensure all these elements in the activations are non-negative, we apply the rectified linear units (ReLU) to the last layer.

As mentioned in Section 2.2, the R-MAC pooling method produces R different regions for each feature map and then calculates MAC features for each region. We represent the region as r∈{1,…,R}. R-MAC uses multiple scale region extraction strategy to take full advantage of the convolutional layer activation information, which is different from MAC. The R-MAC performs MAC pooling on each R to produce an R-dimensional vector for each feature map. Then, all these feature vectors are encoded into a matrix of R×C for the activations. After the operation of normalization, the matrix is concatenated to obtain a C-dimensional feature vector, which could be denoted as $f=\left\{f_{1},\ldots ,f_{C}\right\}$. However, the feature vector does not make full use of the information of each region, because the distribution differs in feature regions. We propose to calculate FDE for feature maps that can reflect the difference in the distribution of pixel values of feature maps in different regions. Then, we combine our proposed FDE with R-MAC feature descriptors. To fuse the two parts better, we apply the operation of L2 and power normalization on FDE vector before fusing. Then, the L2 and power normalizations and PCA are performed on the fused feature descriptors to generate the final features, which would be used for retrieval. The distinctiveness of the final features can be enhanced, and this is further used to improve the performance of image retrieval.

### 3.2. Calculation of FDE

In this section, we represent our idea of the proposed FDE to take the difference of regions into consideration. Our proposed FDE could be helpful to generate more discriminative features that contain richer semantic information and simultaneously eliminate the effects of irrelevant background noise. To be specific, we calculate the proposed FDE and then combine FDE with R-MAC features to produce our final discriminative features. Herein, we design an effective scheme to calculate FDE. This scheme is proposed to focus on the different feature values in regions. We show the details of the processes of FDE calculation, as follows. 

Herein, we would like to describe our proposed scheme in detail by taking one feature map as example. At the first step, we analyze statistical information of the feature values in each region. We build a histogram for each region of feature map. For each region of a certain feature map, there is a range of different feature values. We set the number of blocks in the histogram to B. Then, the value range s of each block is computed according to the maximum and minimum value of the current region. The information of distribution histogram on each region is calculated as the following equation:(6)$$h=\left\{h\left(i,s\right)|1\leq i\leq B;X_{r\left(\min\right)}+\left(i-1\right)S\leq s\leq X_{r\left(\min\right)}+is\right\},S=\left(X_{r\left(\max\right)}-X_{r\left(\min\right)}\right)/B$$
where $X_{r\left(max\right)}$ and $X_{r\left(min\right)}$ are the minimum and maximum value in current region, and the value of B is a parameter that denotes the number of blocks; h(i,s) denotes the statistical values in current region r. The total number of ranges of each region is counted by histogram. 

Probability distribution entropy is an effective method to measure the distribution information of feature maps. Herein, we utilize the distributional probability to compute probability distribution entropy for all these regions. The distributional probability of each region can be calculated by following formula:(7)$$P_{i}=h\left(i,s\right)/\sum_{i=1}^{B}h\left(i,s\right)$$
where h(i,s) denotes the statistical values i-th block. $P_{i}$ is the distributional probability of i-th block in in current region r.

After that, according to the distributional probability of each region that computes the statistical values of distribution histogram h, the probability distribution entropy of each region is calculated using the following formulation:(8)$$H_{r}=-\sum_{i=1}^{B}P_{i}\log P_{i}$$
where $P_{i}$ denotes the distributional probability of block i. 

The probability distribution entropy of a certain region computed by our proposed scheme reflects the distribution of the pixel values of the feature regions. The FDE measures the distribution of feature values in the feature maps. The more concentrated the feature values of the feature maps in a certain region, the smaller the entropy value, and vice versa, as shown in Figure 1. This can reflect the distribution of information in different regions and make descriptors more distinguished. It could be significant to focus on areas that are more useful and eliminate the influence of useless background noises.

As is shown in Figure 1, we assume the large square is a certain feature map generated by the convolutional layer and there are four different feature values within the first region and two different feature values within the second region. The dots of different colors represent feature values in different ranges. The largest value is represented as a blue dot. It is obvious that the distribution in the two regions is inconsistent, but when using the R-MAC algorithm, the MAC feature of the two regions are both 110. R-MAC does not highlight the difference in information distribution between the two regions. We utilize our proposed scheme 1 to calculate the distribution entropy, which could be applied to reflect the difference of the two regions as the values of FDE are different. Then, computed FDE is combined with R-MAC features to improve the effectiveness of image retrieval.

Herein, we discuss the impact of different noises to our proposed FDE scheme. We conduct a set of comparative experiments on computing the values of FDE for a set of 50 images distilled randomly from Oxford5k dataset and after applying four kinds of noises on these images. We give the results in Figure 2. We could make a conclusion as follows. The values of FDE calculated by our proposed method are hardly sensitive to poisson, salt, and speckle noises. Gaussian noise might increase the value of FDE slimly in a very small number of images. We could conclude that our proposed FDE is robust to the most types of noise.

### 3.3. Fusing R-MAC Features with FDE

We aim to use FDE to reflect distribution information of feature regions to overcome the shortcomings of R-MAC. However, the format of FDE is quite different from R-MAC features. It is cautious for us to combine the FDE with R-MAC features. We give more detailed description of the combination schemes, as follows.

The simplest fusing method is to directly concatenate entropy with R-MAC features. There are two ways to concatenate them together. One is to concatenate the regional FDE with R-MAC features directly, which is represented as strategy 1. This strategy will increase the dimension of regional features from R×C to 2R×C. There is another way to fuse them, which would be called strategy 2. The main idea of strategy 2 is to sum the FDE for the whole feature map and then concatenate FDE vector with R-MAC features. The dimension would be increased to (R+1)×C. The two strategies will both increase the dimension of feature vector, which could further increase computational overhead in the process of image retrieval. We design strategy 3, which adopts the weighted summation to fuse the R-MAC features with the FDE without increasing the feature dimension. In strategy 3, the final feature vectors could be obtained by adjusting the weight parameter α. The experiment’s results indicate that strategy 3 will induce better performance of image retrieval. We describe strategy 3 in detail, as follows.

In order to reduce the difference of R-MAC features and FDE, we perform an operation of L2 and power normalization on them separately. We define $p_{1}$ and $p_{2}$ as the power normalization parameter for R-MAC and FDE, respectively. After the normalization operation, the two parts are weighted summed together. The regional feature, which is produced by MAC pooling, is fused with the FDE for a certain feature map by weighted summation as follows:(9)$$f_{c}={\left[f_{c1},\ldots ,f_{cr},\ldots ,f_{cR}\right]}^{T},f_{cr}=f_{cr}^{m}+\alpha H_{cr}$$
where $f_{c}$ denotes the regional feature vector of with FDE of the c-th channel and $f_{cr}$ is the element in the regional feature vector of $f_{c}$, which is weighted summed with an element of R-MAC regional feature $f_{cr}^{m}$ and an element of regional FDE $H_{cr}$ of one feature map. α is an adjustable parameter. We set α=0.5 according to analysis in Section 3.4.

Then, we concatenate all these regional features to generate a feature vector. The concatenation of the element $f_{cr}$ in the vector $f_{c}$ yields the feature descriptors $f_{cE}$ of each channel, and is calculated by the following formula:(10)$$f_{E}={\left[f_{1E},\ldots ,f_{cE},\ldots ,f_{CE}\right]}^{T},f_{cE}=\sum_{r=1}^{R}f_{cr}$$
where $f_{cE}$ is computed by adding all these regional features

Lastly, we perform L2, power normalization, and PCA to generate more effective feature representations, which is significant to improve the performance of image retrieval. We define $p_{3}$ as the power normalization parameter for the fused features. The retrieval process with our fused feature descriptor is illustrated in Figure 3.

As is shown in Figure 3, we follow the deep framework to perform image retrieval. Firstly, we utilize the pre-trained network to extract feature maps. Then, we utilize the R-MAC pooling and FDE scheme mentioned in Section 3.2 to produce deep features for all regions. Then, we fuse them after the process of L2 and power normalization and add up all these regional fused vectors in one feature map to get compact vectors by using the strategy 3 mentioned in Section 3.3. Then, we obtain the final feature descriptors after the process of L2 and power normalization, and we apply the dimensionality reduction of PCA on the final compact feature descriptors, which are then used to perform image retrieval. The experiment results described in Section 4 demonstrate the effectiveness of our proposed method.

### 3.4. Parameter Analysis

In this section, we analyze the main parameters of our algorithm on Paris6k dataset [42] and measure the retrieval accuracy by mean average precision (mAP) [27] as the same to R-MAC. mAP is a commonly evaluation method to measure the performance of image retrieval. It is defined as follows: During the testing phase, we rank all these samples in the testing database by computing their Euclidean distance with the query. Then, we utilize the ranked list to get the average precision (AP) for each query image. Then, we use the AP to compute mAP as follows.
(11)$$mAP=\frac{1}{\left|Q\right|}\sum_{i=1}^{\left|Q\right|}\frac{1}{n_{i}}\sum_{j=1}^{n_{i}}pr\left(i,j\right),pr\left(i,j\right)=\frac{k}{n}$$
where |Q| is the amount of the query images in the dataset, $n_{i}$ is the volume of images in testing dataset which is relevant to i-th query image, Pr(i,j) is the precision of j-th retrieved image to i-th query image, k is the result images returned relevant to the query image, and n is the volume of returned images during retrieving.

In FDE computation, the number of blocks B in the histogram is an adjustable parameter. When fusing the R-MAC features with FDE, the weight α will affect the descriptive ability of the finally generated feature descriptors. In order to fuse the R-MAC features with the FDE, L2 and power normalization processing is required to make them range on the same level. After fusing, we perform a power normalization and L2 normalization again to facilitate subsequent training of PCA. As mentioned in Section 3.1 during the stage of testing, we use multiscale strategy to extract features. We conduct the experiment with different scale size L. And we would give the analysis as follows.

As mentioned in Section 3.2, the number of blocks B plays an important role in computing the FDE. We conduct the experiments on different B with range from 2 to 275 on AlexNet and we set α to 0.5 and $p_{1},p_{2},p_{3}$ to 1.0, 1.1, and 1.1. The results are presented in Table 1.

From Table 1 we can learn that when the value of B becomes larger, the result gradually decreases. The mAP reaches the maximum with 75.01% when B is equal to 2, and we set B to 2 in the later experiments.

Factor α is a parameter used to fuse the computed FDE and R-MAC features. We perform the experiments with α from 0.2 to 1.2 on AlexNet and set B to 2, $p_{1},p_{2},p_{3}$ to 1.0, 1.1, and 1.1, respectively. The results are shown in Table 2. 

From Table 2, we can know that the mAP (%) obtains the maximum value when α=0.5, and the maximum mAP value is bolded. According to the data in the table, α=0.5 is finally selected for the following experiments.

As described in Section 2.3, we utilize L2 and power normalization on R-MAC features and FDE. We conduct the following experiment to find the best values for power normalization. We conduct the experiments with different values from 0.2 to 2 on $p_{1},p_{2}$, and $p_{3}$ separately with α=0.5 and B=2. Then, we show the results in Figure 4. 

We could conclude that the three curves are convex function, which increase and then decrease monotonously. Figure 4 shows that $p_{3}$ is most sensitive to the values of mAP. The maximum values are obtained at 1.0, 1.1, and 1.1, respectively. We set the $p_{1},p_{2},p_{3}$ equal to 1.0, 1.1, and 1.1, respectively, in the rest of the experiments.

We conducted an analysis about different scale size to choose the most appropriate value for L. We performed the following experiment with different scale size for L=1,2,3,4 on AlexNet and VGG16 separately. E indicates feature distribution entropy

We can conclude from Table 3 that when L is smaller than 3, the mAP (%) will increase with the increase of L. However, when L is larger than 3, the mAP (%) will decrease with the increase of L. The best result is obtained when L=3. We set the scale size L to 3 for the following experiments.

## 4. Experiments and Evaluation

In this chapter, we discuss the implementation details and evaluate our algorithm on different datasets. More details are shown as follows.

### 4.1. The Details of Implementation

All our experiments are implemented on Ubuntu 16.04 with a GPU NVIDIA TITAN X and memory of GPU is 64 GB. We use the deep learning tool MatConvNet [43] to realize the convolutional neural network. The experiments use AlexNet [44] and VGG16 [11] pre-trained on ImageNet [45]. We also finetune the network to achieve better performance. We use stochastic gradient descent (SGD) to train Alexnet and Adam to train VGG16. We initialize the learning rate with $l_{0}=10^{-3}$ for SGD and $l_{0}=10^{-6}$ for Adam. The channel of convolutional activation is 256 on AlexNet and 512 on VGG16. We set the input image of resolution to 1024×768. The cosine similarity is used to measure the similarity between the features. The experiments are performed on the Oxford5k [46] and Paris6k [42], Holidays [46], Oxford105k [27], and Paris106k [42]. We utilize mAP to measure the performance of image retrieval.

We conduct our experiments on the following benchmark datasets frequently used for image retrieval. Herein, we give the details of these datasets as follows.

Oxford5k [27] is a dataset provided by Flickr with a total of 5062 images. It contains 11 different landmarks of Oxford. Oxford5k [27] owns five query areas for each landmark building. Each image is labeled as one of four tags: Good, okay, junk, bad. The first two match the current query area; not good means that the error is matched.

Paris6k [42] is usually used in conjunction with Oxford5k [27] and it is also provided by Flickr with 11 classes. Each class has five query areas with a total of 6412 images about Paris buildings. It is also labeled with a four-category label, which is similar to Oxford.

Flickr100k [27] is made up of 1,000,071 high-resolution images from 145 of the most popular tags on Flickr, and is late added to the Oxford5k and Paris6k to become Oxford105k [27] and Paris106k [42] for large-scale image retrieval.

Holidays [46] mainly contains a variety of landscape pictures. It consists of 1491 images with 500 groups of similar images; each group has a query image. Unlike Oxford5k [27] and Paris6k [42], the query image on Holidays is the entire image rather than the region of interest (ROI).

### 4.2. The Calculation and Fusing Schemes of Entropy

In this section, we analyze the effect of different fusion strategies. As mentioned in Section 3.3, we show three strategies to fuse feature distribution entropy with R-MAC features. Strategy 1 stitches the region entropy directly to R-MAC features of each region. Strategy 2 is to stitch the entropy of the entire feature map to the R-MAC feature. Strategy 3 uses weighted summation. We use the pre-trained AlexNet and VGG16 to perform a series of comparative experiment with the three strategies on Paris6k, respectively. The results are shown in Table 4.

From the results shown in Table 4, we can make the following conclusions. The result of strategy 3 achieves 75.01% and is better than other fusion schemes when the experiment is conducted on AlexNet. Then, we perform the experiment of the three schemes on VGG16. We find that scheme 3 still gains the best result and obtains 83.50%. It can be seen from Table 4 that both scheme 2 and scheme 3 have been improved, but scheme 3 is more effective, and scheme 3 is adopted for the following experiments.

### 4.3. Compact Representation Comparison

In this section, we conduct the experiments with our proposed method and verify the compatibility of our proposed algorithm. To gain more, we perform PCA whitening to reduce the influence of noise with no dimensionality reduction. We use MAC, R-MAC, and our proposed method for the comparative experiments; the calculation and fusion method of FDE is selected in Section 4.1. 

PCA whitening is one of the most important post-processing methods, which can reduce noise influence and improve retrieval efficiency. In order to verify the impact of PCA whitening on the retrieval results, we conducted comparative experiments on AlexNet [10] and VGG16 [47] on Paris6k [42], Oxford5k [27], and Holidays [46]. The results are shown in Table 5.

Table 5 shows the results of different feature descriptors before and after PCA whitening. It should be noted that whether using PCA whitening or not, the feature dimension in AlexNet is 256, the same as VGG16 is 512. We train PCA on Oxford5k and then use it to test on Holidays or Paris6k and similarly we train PCA on Paris6k to test on Oxfor5k. From this table we can learn that when we test on AlexNet, R-MAC+E+P on Paris6k, Oxford6k, and Holidays achieve 75.01%, 57.15%, and 82.76%, respectively, and the best retrieval results are obtained. When we perform the experiments on VGG16, the best results are 83.56%, 57.58%, and 86.90% on Paris6k, Oxford5k, and Holidays, respectively. We can also see that whether it is MAC, R-MAC, or R-MAC+E, the mAP value of using PCA compared with the one without using PCA has been significantly improved in most cases, with only a slight drop in the MAC method using AlexNet on Holidays. It is fully proved that PCA whitening can effectively improve retrieval performance in most cases.

Fusion representations. In order to verify the compatibility of the algorithm, we designed four sets of comparative experiments, using pre-trained networks on VGG16 and AlexNet to perform experiments. Oxford5k and Paris6k use the query area specified in the 55 query images given by the datasets. We compare the MAC and R-MAC features with features after fusing entropy by mAP (%). The results are shown in Table 6.

The conclusions drawn from Table 6 are as follows. The results in the table indicate that when we experiment on AlexNet, R-MAC+E obtains the best results on all these datasets with 57.15%, 50.19%, 75.01%, 63.29%, and 82.76%, which has been bolded. The same conclusion is obtained when we perform the experiments on VGG16. We get the results of 69.64%, 64.91%, 83.56%, 77.89%, and 86.90%, which are the maximum values on the different datasets. We have bolded these maximum values in Table 6. We can know that the fused feature representations generated by using our proposed method could be more effective and gain a better performance.

Re-ranking and QE. As mentioned in Section 2.4, re-ranking and QE can further improve the performance of image retrieval. In this section, we use the pooling methods above to calculate feature descriptors. Then, we examine the advantage of re-ranking and QE on Oxford5k, Paris6k, Oxford105k, and Paris106k. We show the results in Table 7.

From Table 7, we could know that the best results are achieved when we apply re-ranking and QE on our proposed method with the mAP (%) being 67.83%, 61.01%, 81.04%, and 70.75% on the four different datasets on AlexNet. We get the same conclusion on VGG16 with the maximum values of mAP (%) being 78.48%, 75.79%, 86.53%, and 81.22%. We could conclude from Table 7 that the results of the features would likely increase when we apply the operation of re-ranking and QE on four different datasets on AlexNet and VGG16.

Comparison with state-of-the-art algorithms. In order to demonstrate the effectiveness and superiority of our algorithm, our experimental results are compared with other state-of-the-art algorithms. We conduct experiments not only with raw image representations, but also use representations performed with re-ranking and query expansion. The results can be seen in Table 8. 

We can see from Table 8 that our proposed method obtains the best results in most cases. After performing re-ranking and query expansion, the performance has been significantly improved. According to the experimental data, our algorithm after re-ranking and query expansion achieves the best results in almost all categories, which fully demonstrates the effectiveness and superiority of the improved algorithm.

Test on fine-tuned network. The network proposed by Filip et al. [25] uses the contrastive loss function to train the parameters in the network. In order to adapt the network parameters to our algorithm and achieve better performance, four sets of comparative experiments were performed on Oxford5k, Oxford105k, Paris6k, and Paris106k and Holidays. The results are shown in Table 9.

We can make the following conclusion from the results achieved on fine-tuned network. When we experiment on AlexNet, MAC+E gains the best results on Oxford5k and Oxford105k. R-MAC+E achieve the best results on Paris6k, Paris106k, and Holidays. When we test on the fine-tuning network initialized with VGG16, the best results were obtained on MAC+E on Oxford5k, Oxford105k, Paris6k, and Paris106k, and on R-MAC+E on Holidays. The experimental results indicate that the feature distribution entropy can also be used in the fine-tuned network to promote the performance of image retrieval.

Experiment on medical dataset. To further demonstrate our proposed method, we conduct a set of experiments on medical dataset with the methods of MAC, MAC+E, R-MAC, and R-MAC+E on the AlexNet, which is pre-trained on ImageNet. The dataset for performing our experiments is composed of two public medical datasets of Brain_Tumor_Dataset [48] and Origa [49]. We present the results in Table 10.

We can learn from Table 10 that our proposed method of fusing our FDE with R-MAC features obtains the best result with mAP is 92.93%, which is higher than R-MAC by 0.04%. The mAP of fusing MAC with our FDE would be increased by nearly 4.5% compared to MAC. The results in Table 10 shows that our proposed FDE is effective in improving the performance of image retrieval. Furthermore, the results demonstrate that our proposed method of fusing FDE with R-MAC outperforms the existing methods.

### 4.4. Discussion

We would like to give some discussion for our proposed method. Taking the distribution information in different regions into consideration and combining the distribution information with R-MAC features could obtain remarkable performance in image retrieval. We design an effective scheme to calculate FDE, which is significant for promoting the performance of image retrieval. Here, we propose a superior strategy of weighted summation to fuse our proposed FDE with R-MAC feature descriptors to generate more informative feature representations. Furthermore, the post-processing of re-ranking and QE would be helpful to promote the effectiveness of our proposed method. When we test the five public datasets on AlexNet, our method can achieve state-of-the-art performance in image retrieval. When we test on VGG 16, we obtain the best results for most datasets and acceptable results on Oxford5k lower than BoW-CNN.

## 5. Conclusions

In this paper, we proposed to make full use of the regional distribution information to generate more informative feature representations to promote the performance of image retrieval. We proposed to utilize FDE to reflect the difference of distribution information in different regions. We designed an effective scheme to calculate our proposed FDE, and the experimental results show that our FDE is effective to improve the performance of image retrieval. Then, we proposed a superior strategy to fuse the proposed FDE with R-MAC features to generate more effective deep representations, which could achieve prominent performance in the task of image retrieval. In order to demonstrate the compatibility of our proposed method, we also conducted the experiments with the fused features on different datasets on the pre-trained network. The results show that the performance with our proposed method outperforms that of the existing state-of-the-art methods. Furthermore, we used the post-process methods of re-ranking and QE to further improve the performance. Finally, we used the fine-tuned network and medical dataset to verify the effectiveness of our proposed method. We obtained state-of-the-art results with our proposed method on five different datasets.

Our method mainly focuses on how to make full use of the feature maps output from the pre-trained network. We would like to pay attention to train more suitable network for the task of image retrieval, and we would concentrate on improving the effectiveness and robust of our network by designing more effective loss function and network architecture in our following works.

## Figures and Tables

**Figure 1 entropy-21-01037-f001:**
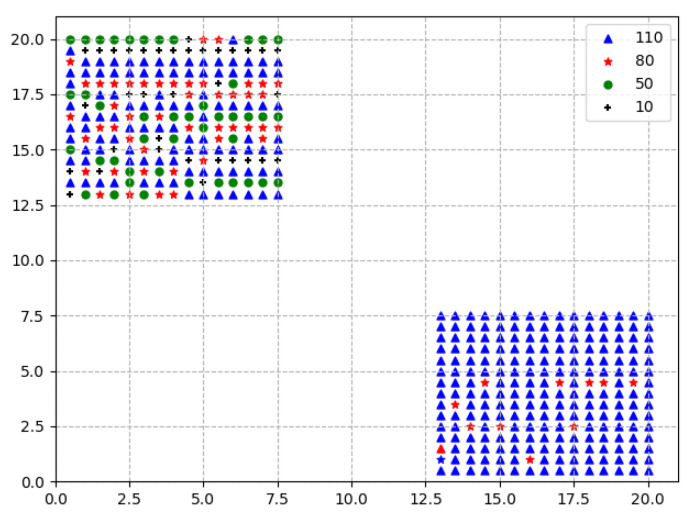
The two regions with different distribution in a feature map. The x-axis and y-axis denote the position of the feature values. The dots with different colors are the feature values. To be specific, the blue, red, green, and black dots are the values of 110, 80, 50, and 10, respectively.

**Figure 2 entropy-21-01037-f002:**
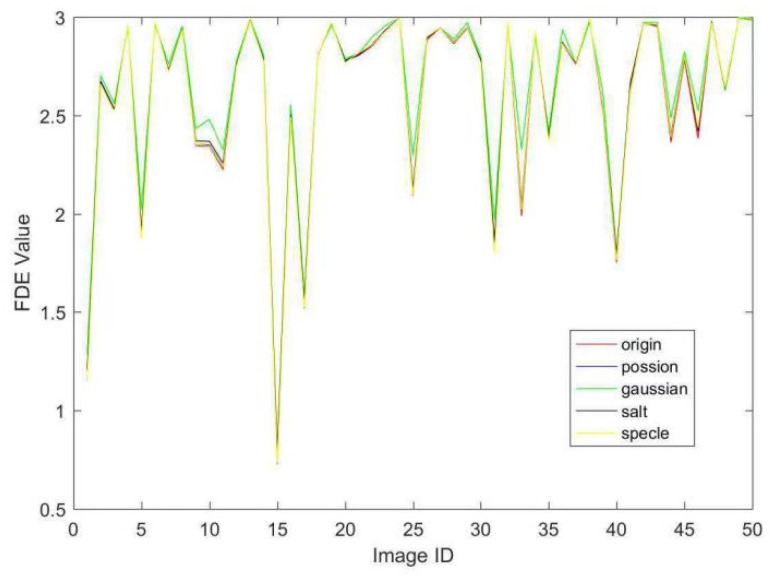
The feature distribution entropy (FDE) curve of 50 images applied four kinds of noise. The x-axis represents the image ID and y-axis denotes the values of FDE. The red curve is the FDE values of original images. The other four curves denote the FDE values of the same images with Poisson, Gaussian, salt, and speckle noise, respectively.

**Figure 3 entropy-21-01037-f003:**
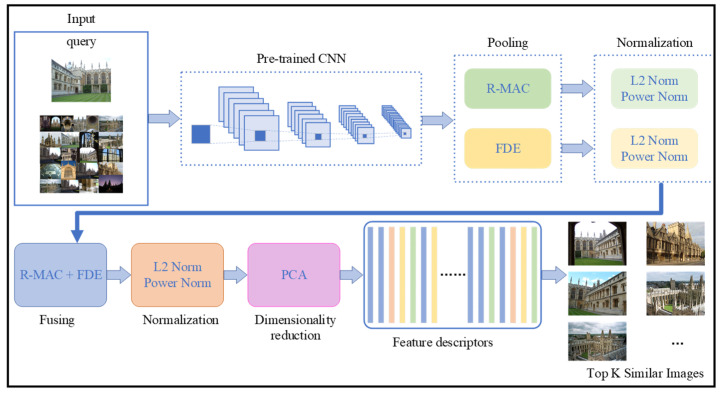
The process of image retrieval with our proposed fused features. The pre-trained convolutional neural network (CNN) is used to produce feature maps on the input query image and a retrieval set. The region maximum activations of convolutions (R-MAC) pooling and the proposed FDE are applied on feature maps and the normalization is applied on them separately. Then, weighted fusion strategy is utilized to produce raw features. We finally perform normalization and principal component analysis (PCA) on the fused features to generate the final feature descriptors, which are used in the following retrieval stage.

**Figure 4 entropy-21-01037-f004:**
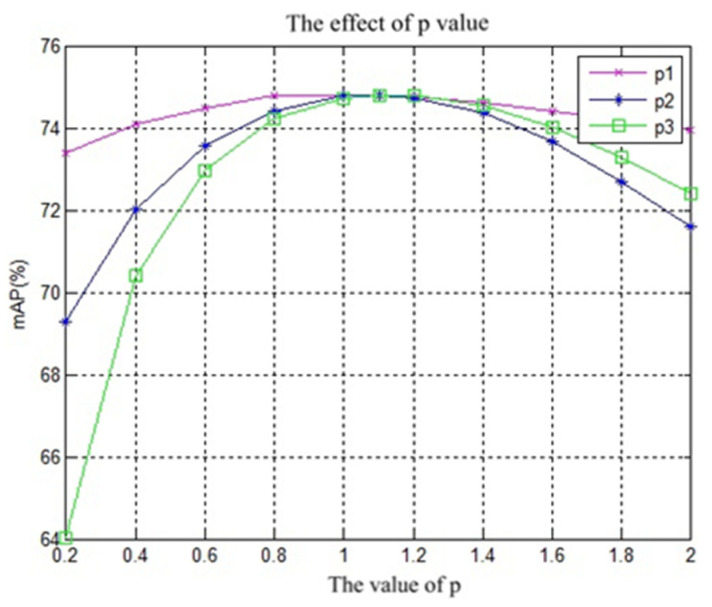
The changes of mean average precision (mAP) with $p_{1},p_{2},p_{3}$. The x-axis is the value of p, which could be used to denote the values of $p_{1},p_{2},p_{3}$. The y-axis is the value of mAP (%).

**Table 1 entropy-21-01037-t001:** The influence of B on retrieval.

.B	2	15	50	100	175	275
**mAP(%)**	**75.01**	74.44	74.21	74.12	74.06	74.03

**Table 2 entropy-21-01037-t002:** The influence of α on retrieval.

ɑ	0.2	0.4	0.5	0.6	0.8	1.0	1.2
**mAP(%)**	73.69	73.95	**74.05**	73.95	73.80	73.58	73.31

**Table 3 entropy-21-01037-t003:** The results with different scale size.

Methods.	AlexNet				VGG16			
**L**	**1**	**2**	**3**	**4**	**1**	2	3	4
**R-MAC**	47.9	54.6	56.1	55.6	57.3	64.5	66.9	67.44
**R-MAC+E**	54.00	57.03	**57.15**	56.38	64.04	67.97	**69.64**	69.09
**MAC**	44.83				55.01			

**Table 4 entropy-21-01037-t004:** The strategies of fusing with R-MAC features.

Network	Fusing	mAP(%)
**AlexNet**	1	71.15
	2	73.56
	3	**75.01**
	R-MAC	72.95
**VGG16**	1	79.46
	2	83.02
	3	**83.50**
	R-MAC	83.02

**Table 5 entropy-21-01037-t005:** Performance (mAP) comparison with or without PCA whitening. P: Performing PCA whitening (+P), E: Fusing with feature distribution entropy (+E), or without entropy. The best result is highlighted in bold.

Pooling.		AlexNet			VGG16	
	Paris6k	Oxford5k	Holidays	Paris6k	Oxford5k	Holiday
**MAC+P**	54.42	44.83	68.75	74.73	55.01	75.23
**R-MAC**	66.82	50.99	75.75	75.31	56.23	81.26
**R-MAC+P**	72.95	56.06	80.99	83.02	66.71	84.04
**R-MAC+E**	66.35	49.71	76.90	74.36	57.58	81.66
**R-MAC+E+P**	**75.01**	**57.15**	**82.76**	**83.56**	**69.64**	**86.90**

**Table 6 entropy-21-01037-t006:** Performance (mAP (%)) comparison between fusing with FDE or without. E: Fusing with FDE (+E), or without entropy. The best result is highlighted in bold.

Network.	Pooling	Oxford5k	Oxford105k	Paris6k	Paris106k	Holidays
**AlexNet**	MAC	44.83	34.84	54.42	37.09	68.75
	MAC+E	51.73	45.54	64.78	50.34	78.01
	R-MAC	56.06	46.85	72.95	60.07	80.99
	R-MAC+E	**57.15**	**50.19**	**75.01**	**63.29**	**82.76**
**VGG16**	MAC	55.01	48.50	74.73	62.46	75.23
	MAC+E	62.57	58.75	79.51	72.57	82.07
	R-MAC	66.71	62.35	83.02	76.28	84.04
	R-MAC+E	**69.64**	**64.91**	**83.56**	**77.89**	**86.90**

**Table 7 entropy-21-01037-t007:** Performance (mAP(%)) comparison using re-ranking and query expansion or without, R: Using re-ranking (+R), QE: Using query expansion (+QE). E: Fusing with feature distribution entropy (+E), or without entropy. The best result is highlighted in bold.

Network	Pooling	Oxford5k	Oxford105k	Paris6k	Paris106k
**AlexNet**	MAC+R	59.01	46.26	65.67	45.61
	MAC+R+QE	63.92	50.21	69.13	49.08
	MAC+E+R	64.02	54.03	73.95	58.51
	MAC+E+R+QE	70.46	59.67	76.60	61.16
	R-MAC+R	61.13	55.16	77.52	65.63
	R-MAC+R+QE	66.85	60.68	80.42	69.02
	R-MAC+E+R	62.37	56.46	78.35	67.49
	R-MAC+E+R+QE	**67.83**	**61.01**	**81.04**	**70.75**
**VGG16**	MAC+R	70.57	60.39	81.16	64.29
	MAC+R+QE	74.21	63.65	82.84	69.01
	MAC+E+R	76.09	67.97	84.36	76.37
	MAC+E+R+QE	78.95	71.47	85.22	76.99
	R-MAC+R	74.54	70.89	85.16	79.29
	R-MAC+R+QE	77.33	74.69	86.45	80.73
	R-MAC+E+R	75.96	72.97	85.33	79.97
	R-MAC+E+R+QE	**78.48**	**75.79**	**86.53**	**81.22**

**Table 8 entropy-21-01037-t008:** Performance (mAP (%)) comparison with the state-of-the-art algorithms. Dim: Dimensionality of final compact image feature descriptors, Not Applicable (N/A) for the bag-of-visual-words (BoW)-CNN due to its sparse representations. R: Using re-ranking (+R), QE: Using QE (+QE). E: Fusing with feature distribution entropy (+E), or without entropy. The best result is highlighted in bold.

Network	Pooling	Dim	Oxford5k	Oxford105k	Paris6k	Paris106k	Holidays
**Original retrieval results**							
**AlexNet**	MAC [12]	256	44.24	34.84	54.42	37.09	68.75
	R-MAC [24]	256	56.06	46.85	72.95	60.07	80.99
	R-MAC+E	256	**57.15**	**50.19**	**75.01**	**63.29**	**82.76**
**VGG16**	SPOC [15]	256	53.1	50.1	-	-	80.2
	uCrow [23]	256	66.7	61.2	73.9	65.8	81.5
	MFC [32]	256	68.4	62.9	83.4	-	-
	MAC [17]	512	55.01		74.73		75.23
	SPOC [15]	512	56.4	47.8	72.3	58.0	79.0
	uCrow [23]	512	69.7	64.1	78.6	71.0	83.9
	BoW-CNN [26]	N/A	73.9	59.3	82.0	64.8	-
	NetVLAD [16]	4096	55.5	-	67.7	-	82.1
	MFC [32]	512	70.6	65.3	83.3	-	-
	R-MAC [24]	512	66.71	62.35	83.02	76.28	84.04
	R-MAC+E	512	**69.64**	**64.91**	**83.56**	**77.89**	**85.90**
**After re-ranking (R) and query expansion (QE)**							
**AlexNet**	MAC+R+QE [12]	256	63.92	50.21	69.13	49.08	-
	R-MAC+R+QE [24]	256	66.85	60.68	80.42	69.02	-
	R-MAC+E+R+QE	256	**67.83**	**61.01**	**81.04**	**70.75**	-
**VGG16**	Crow+QE [23]	512	74.9	70.6	84.8	79.4	-
	MAC+R+QE [12]	512	74.21	63.65	82.84	69.01	-
	R-MAC+R+QE [24]	512	77.33	74.69	86.45	80.73	-
	BoW-CNN+R+QE [26]	512	**78.8**	65.1	84.8	64.1	**-**
	R-MAC+E+R+QE	512	78.48	**75.79**	**86.53**	**81.22**	**-**

**Table 9 entropy-21-01037-t009:** The results of image retrieval on five different datasets with fine-tuned network on AlexNet and VGG16. R: Using re-ranking (+R), QE: Using query expansion (+QE). E: Fusing with feature distribution entropy (+E), or without entropy.

Network	Pooling	Oxford5k	Oxford105k	Paris6k	Paris106k	Holidays
**AlexNet**	MAC	61.20	50.29	68.25	53.40	73.42
	MAC+E	**66.57**	**58.14**	74.13	61.61	78.95
	R-MAC	63.68	53.12	73.35	60.02	78.96
	R-MAC+E	64.96	54.96	**75.79**	**63.01**	**79.72**
**VGG16**	MAC	81.34	75.29	83.90	75.22	80.11
	MAC+E	**84.23**	**78.65**	**86.80**	**80.36**	82.25
	R-MAC	80.73	72.67	85.08	77.64	82.43
	R-MAC+E	81.91	73.82	85.91	79.06	**83.39**

**Table 10 entropy-21-01037-t010:** Performance (mAP (%)) comparison between fusing with FDE or without. E on medical dataset: Fusing with FDE (+E), or without entropy. The best result is highlighted in bold.

Pooling	mAP(%)
**MAC**	86.64
**MAC+E**	91.11
**R-MAC**	92.89
**R-MAC+E**	**92.93**

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
