# Peer review of "Fusing Feature Distribution Entropy with R-MAC Features in Image Retrieval"

_entropy, 2019, doi:10.3390/e21111037_

Round 1

Reviewer 1 Report

The authors propose an improvement of R-MAC pooling method using feature distribution entropy. They proposed a method to fuse the entropy with the R-MAC features, without increasing the dimensions of the feature vector. The method is tested using the databases used for similar testing in numerous papers proposing different improvements in image retrieval.

The authors recycle their idea to use Shannon entropy. They have already implemented this idea in the Entropy journal (but not in the context of CNN), [A1] Qiuzhan Zhou, Cheng Wang, Pingping Liu, Qingliang Li, Yeran Wang and Shuozhang Chen: “Distribution Entropy Boosted VLAD for Image Retrieval”, and also in the paper [21] (again, not in the context of CNN) Liu, P.; Zhuang, M.; Guo, H.; Wang, Y.; Ni, A.: “Adding spatial distribution clue to aggregated vector in image retrieval”. Eurasip Journal on Image & Video Processing 2018.

The paper is very difficult to read, due to the three drawbacks:

The parts of the paper seem not to be well connected one to another, as if taken from somewhere else, and the abbreviations are not explained (while in similar other papers, the same abbreviations are explained). Most of the important issues are not clarified, while some basic ones (e.g. how to calculate the histogram) are described in detail. English does not seem to be correct even to a non-native English speaker, and there are typographical errors, even considering the formulae.

These issues are elaborated in detail as follows:

Issues 1 and 2 (jointly)

The sections Introduction and Related work are both presenting an overview of the existing methods. But numerous papers, both conference and journal, can be used as a template for this introductory talk, e.g. [A2] Magliani, Prati: “An accurate retrieval through R-MAC + descriptors for landmark recognition”.

Lines 99-120 are devoted to the Section entitled “2.2. Entropy for image retrieval”. But entropy (the previous contribution of the authors) is mentioned in a single sentence in lines 108-109, so it is unclear why the complete subsection is named “Entropy”.

In the introduction, some unknown parameters have been mentioned, K*D*1. What is K, what is D, what are the dimensions of 1? Does “*” mean “multiply” or “convolve” or something else?

Line 69, small “p” in “pooling methods”.

Section 3.1. “Algorithm proposed background”. Seemingly, it means “The background of the proposed algorithm”. R-MAC is explained without defining the variables. It is said that L can be 1, 2, 3, but then L goes from 1 to m (what is m?), min(W,H) is mentioned – what is W, what is H, then it is written “R-MAC uniformly extract ? ∗ (? + ? + 1) regions that Edges equal (?,?)/(? + 1) at each scale” what is (small) l, why Edged with capital E? The number of features C in R*C is not defined. The same subsection in e.g. [A2] is written quite clearly and clearly defines all the variables.

Considering the background of the proposed algorithm, some of the procedures are not defined, i.e. power normalization and L2 normalization used in 3.5. and at least a couple of sentences should be devoted to PCA.

“Our work is to improve retrieval performance through re-ranking and QE.” So the paper should describe re-ranking and QE.

Section 3.2. “Feature distribution entropy” is not clarified. “We calculate the probability of the distribution … and substitute into Eq. (1)”. But Eq. 1 is Shannon’s formula for the entropy of the discrete source (it also applies to the probability density functions estimated using the histograms) with statistically independent symbols. What is “probability of the distribution”? If it concerns a distribution function, each its value is a probability itself (F(x0) = Pr{x>x0}). But since it is Shannon’s formula, it should be a probability density f(x) estimated using a histogram. Then, why small “p” in Eq. 1, and capital “P” in line 167 below the Eq. 1? Besides, Shannon formula is well known from basic courses in engineering, in the 2nd and the 3rd years of study, and it is not needed to be repeated for the second time in Eq 4.

Equations 2 and 3 explain in detail how to estimate a histogram which would be OK had the authors described other features of their contribution in such details. Then Eq. 4 repeats Eq 1.

But it is not clear what is counted in the histograms “Each region has its own range of feature pixel values”. Figure 1 is not helpful, it may be assumed that there are only four different feature values within the first region and two different feature values within the second region. But it is not clear how to get the feature values.

ReLU layer is mentioned but not defined.

Taking zeros considering entropy is another problem, but should not be the scope of this paper. However, it is said later (Section 4.1.) that it has a different fusion procedure?

Section 3.3.

It is not described, for Eq 5, how to chose alpha. Then (as already said), it is too laconically to say “then perform normalization and PCA”.

Section 3.4. is not necessary: pseudocode does not bring more information than the text explanation in Sections 3.2. and 3.3.

Section 3.5. A parameter p appears, line 267: “Figure 2 shows the influence of parameter ?, assumed that the power normalization parameter after calculating the entropy is ?1, the power normalization parameter after calculating the R-MAC is ?2, and after fusing is set to ?3.” What is parameter p, and how do you obtain p1, p2, and p3?

It should be nice if mAP is defined since it is the basic measure for retrieval accuracy.  

Table 1 concludes that a binary histogram should be used? And what was the parameter alpha when calculating mAP for Table 1?

In lines 276-278 parameter “a” is mentioned, while in Table 2 it is “alpha”. And what B values were used for calculations in Table 2?

In Section 4.1, Section 3.3, equations 5 and 6, explain the fusion, but Section 4.1. states “there are many ways to fuse feature distribution entropy with R-MAC features.” Then a Scheme 3, which was not mentioned before, appears and both Scheme 2 and Scheme 3 have a fusion of their own.

Section 4.2. presents the results, and the results are discussed in a different and much better way than the methods explained in the previous section was given. The authors performed several experiments using common datasets.  Even the English language is better. The authors only forgot to put the best results in bold (as promised in Captions of Tables 2, 5, 6, 7 and 8.

Issue 3:

Language is a problem, although it improves in Section 4.2. It is almost unreadable in Sections 1 – 4.1. It is not a major problem – their paper [A1] published in Entropy is not in perfect English either.

Some examples:

Authors tend to glue reference and abbreviation, making the text less readable: “retrieval(CBIR)[1-3]”, it should be  “retrieval (CBIR) [1-3]” (many examples)

“we perform re-ranking[19]and query expansion(QE)[22] on …”

“outputted”

“Researchers tend to use the feature maps come from the last convolutional layer.”

“NetVLAD which inspired by VLAD and produces (?∗?)∗? dimentional aggregated descriptors which can be directly used”

In my opinion, the authors should quote the original Shannon paper, as it was a foundation of Information Theory as a scientific branch.

Author Response

On behalf of my co-authors, we thank you very much for giving us an opportunity to revise our manuscript. We appreciate editor and reviewers very much for the positive and constructive comments and suggestions on our manuscript (ID: 611445).

We have studied reviewer’s comments carefully and have made revision. According to your advice, we amended the relevant part in manuscript. Some of your questions were answered as bellow.

1) The authors recycle their idea to use Shannon entropy. They have already implemented this idea in the Entropy journal (but not in the context of CNN), [A1] Qiuzhan Zhou, Cheng Wang, Pingping Liu, Qingliang Li, Yeran Wang and Shuozhang Chen: “Distribution Entropy Boosted VLAD for Image Retrieval”, and also in the paper [21] (again, not in the context of CNN) Liu, P.; Zhuang, M.; Guo, H.; Wang, Y.; Ni, A.: “Adding spatial distribution clue to aggregated vector in image retrieval”. Eurasip Journal on Image & Video Processing 2018.

Answer: We are very grateful to your kind question. This is our first time to apply entropy to CNN-based method, and we have conducted the experiments with our proposed method to verify the effectiveness of our proposed FDE.

2) The parts of the paper seem not to be well connected one to another, as if taken from somewhere else, and the abbreviations are not explained (while in similar other papers, the same abbreviations are explained). Most of the important issues are not clarified, while some basic ones (e.g. how to calculate the histogram) are described in detail. English does not seem to be correct even to a non-native English speaker, and there are typographical errors, even considering the formulae.

Answer: Thanks for your kind advice and comment for our manuscript. We reorganized our paper and refine the architecture of our paper and explain the abbreviations in our paper. Furthermore, we clarify the issues we need to solve in the section 1,2,3 and the process of computing our proposed FDE in section 3. And we have tried our best to improve our language representation in English to enhance the readability of our manuscript.

3) The sections Introduction and Related work are both presenting an overview of the existing methods. But numerous papers, both conference and journal, can be used as a template for this introductory talk, e.g. [A2] Magliani, Prati: “An accurate retrieval through R-MAC + descriptors for landmark recognition”.

Answer: We are very grateful to your kind comments for the manuscript. We revise the introduction and related work by using “An accurate retrieval through R-MAC + descriptors for landmark recognition” as a reference.

4) Lines 99-120 are devoted to the Section entitled “2.2. Entropy for image retrieval”. But entropy (the previous contribution of the authors) is mentioned in a single sentence in lines 108-109, so it is unclear why the complete subsection is named “Entropy”.

Answer: Thanks for your helpful comments. We have renamed the section 2.2 as “Compact features with spatial information”. As section 2.2 is represented to studies of distribution information in features,we attempt to use entropy to describe the distribution information to enhance the discriminative ability of features. The name of “Compact features with spatial information” is more suitable for this section.

5) In the introduction, some unknown parameters have been mentioned, K*D*1. What is K, what is D, what are the dimensions of 1? Does “*” mean “multiply” or “convolve” or something else? Line 69, small “p” in “pooling methods”.

Answer: We are very grateful to your kind comments for the manuscript. We have deleted these unclear descriptions and have revised the unclear representation in the relevant parts.

6) Section 3.1. “Algorithm proposed background”. Seemingly, it means “The background of the proposed algorithm”. R-MAC is explained without defining the variables. It is said that L can be 1, 2, 3, but then L goes from 1 to m (what is m?), min(W,H) is mentioned – what is W, what is H, then it is written “R-MAC uniformly extract ? ∗ (? + ? + 1) regions that Edges equal (?,?)/(? + 1) at each scale” what is (small) l, why Edged with capital E? The number of features C in R*C is not defined. The same subsection in e.g. [A2] is written quite clearly and clearly defines all the variables.

Answer: Thanks for your kind advice and comment for our manuscript. We reorganize this section. We explain the R-MAC in section 2.1 named “Pooling approaches” and specifically in line 117-159 and the variables used has been explained. In section 3.1, we define the variables which could be used in the rest of our paper and describe our work roughly to elicit specific method in the rest part.

7) Considering the background of the proposed algorithm, some of the procedures are not defined, i.e. power normalization and L2 normalization used in 3.5. and at least a couple of sentences should be devoted to PCA.

Answer: We are very grateful to your kind question. We give detailed illustration in section 2.3 which has defined the procedures of power normalization, L2 normalization and PCA. And the reference of PCA is [40], L2 / power Normalization is [13]

8) “Our work is to improve retrieval performance through re-ranking and QE.” so the paper should describe re-ranking and QE.

Answer: Thanks for your kind advice and comment for our manuscript. We deleted this description as improving retrieval performance through re-ranking and QE is not our main contribution, it is a post-process which could be used to further improve the performance of image retrieval in our paper. And we also describe re-ranking and QE in section 2.4 as we utilize them to further improve the performance.

9) Section 3.2. “Feature distribution entropy” is not clarified. “We calculate the probability of the distribution … and substitute into Eq. (1)”. But Eq. 1 is Shannon’s formula for the entropy of the discrete source (it also applies to the probability density functions estimated using the histograms) with statistically independent symbols. What is “probability of the distribution”? If it concerns a distribution function, each its value is a probability itself (F(x0) = Pr{x>x0}). But since it is Shannon’s formula, it should be a probability density f(x) estimated using a histogram. Then, why small “p” in Eq. 1, and capital “P” in line 167 below the Eq. 1? Besides, Shannon formula is well known from basic courses in engineering, in the 2nd and the 3rd years of study, and it is not needed to be repeated for the second time in Eq 4.

Answer: We are very grateful to your kind question. We have deleted these inappropriate descriptions. We utilize the probability of the distribution to compute our FDE. The “probability of the distribution” is used to reflect the feature values distribution of different regions and is computed through the distribution histogram. It has been deleted in the illustration of building the histogram for each region. And it has been described in the part of illustrating “probability distribution entropy”. And we have deleted Eq 1 in this revised manuscript. We also check our distribution histogram equation (Eq 2 in the old manuscript) and make a modification as Eq 6 in the updated manuscript.

10) Equations 2 and 3 explain in detail how to estimate a histogram which would be OK had the authors described other features of their contribution in such details. Then Eq. 4 repeats Eq 1.

Answer: Thanks for your kind advice and comment for our manuscript. We have given details of our main contribution and works in section 3 And we delete Eq 1 in this updated manuscript.

11) But it is not clear what is counted in the histograms “Each region has its own range of feature pixel values”. Figure 1 is not helpful, it may be assumed that there are only four different feature values within the first region and two different feature values within the second region. But it is not clear how to get the feature values.

Answer: We are very grateful to your kind comments and advice. We have re-described figure 1 in section 3.2. We assume there are four different feature values within the first region and two different feature values within the second region. And the feature values are from the feature map which is computed from the convolutional layer.

12) ReLU layer is mentioned but not defined.

Answer: We are very grateful to your kind comments for the manuscript. We have deleted the contents related to the ReLU. This part is to describe scheme 2 in the old manuscript, as it is not important for our main work the performance by using scheme 2 is effective enough to improve the performance of image retrieval.

13)Taking zeros considering entropy is another problem, but should not be the scope of this paper. However, it is said later (Section 4.1.) that it has a different fusion procedure?

Answer: Thanks for your kind advice and comment for our manuscript. We have removed this part of content as it is not important for our main work and the performance by using scheme of calculating the distribution entropy of non-zero values is effective to improve the performance of image retrieval. And the section 4.1 shows the results of fusion strategies (line 454-467) and according to the results we chose scheme 1 to calculate FDE and strategy 3 to fuse FDE with R-MAC features to generate our compact feature representation.

14) Section 3.3. It is not described, for Eq 5, how to choose alpha. Then (as already said), it is too laconically to say “then perform normalization and PCA”.

Answer: We are very grateful to your kind comments. The Eq 5 in former version is Eq 10 in this manuscript.  is a parameter used to combine FDE with R-MAC features, and we chose it according to section 3.2. We clarify it in section 3.3 and 3.4. We have given the introduction of PCA and L2 / power Normalization in section 2.3. And the reference of PCA is [40], L2 / power Normalization is [13]

15) Section 3.4. is not necessary: pseudocode does not bring more information than the text explanation in Sections 3.2. and 3.3.

Answer: Thanks for your kind advice and comment for our manuscript. We have deleted section 3.4 and the pseudocode. We show the process of our method in figure 3 in section 3.3.

16) Section 3.5. A parameter p appears, line 267: “Figure 2 shows the influence of parameter ?, assumed that the power normalization parameter after calculating the entropy is ?1, the power normalization parameter after calculating the R-MAC is ?2, and after fusing is set to ?3.” What is parameter p, and how do you obtain p1, p2, and p3?

Answer: We are very grateful to your kind comments.  is an abstract representation that is a parameter used in the power normalization. As we make power normalization for three times in producing the final feature vector, we need to choose the most appropriate value of  for each power normalization. We use to describe the  value individually for these three times. We conduct the experiments with different values from 0.2 to 2 on  and  separately. And then we analyze the results represented in Figure 4 to obtain the best value of   to achieve the maximum of mAP (%).

17) It should be nice if mAP is defined since it is the basic measure for retrieval accuracy. 

Answer: Thanks for your kind advice and comment for our manuscript. We give the definition of mAP in section 3.4 and specifically in line 379-383.

18) Table 1 concludes that a binary histogram should be used? And what was the parameter alpha when calculating mAP for Table 1?

Answer: We are very grateful to your kind comments. As these parameters are interactional, when we analyze a certain parameter with others is the best one. We set  when we analyze .

19) In lines 276-278 parameter “a” is mentioned, while in Table 2 it is “alpha”. And what B values were used for calculations in Table 2?

Answer: Thanks for your kind comment for our manuscript. We have revised this part. In the revised manuscript, we clarify that   in relevant experiments.

20) In Section 4.1, Section 3.3, equations 5 and 6, explain the fusion, but Section 4.1. states “there are many ways to fuse feature distribution entropy with R-MAC features.” Then a Scheme 3, which was not mentioned before, appears and both Scheme 2 and Scheme 3 have a fusion of their own.

Answer: We are very grateful to your kind comments. We describe other two strategies in section 3.3 of line 329-337 which are represented as strategy 1 and strategy 2 and we describe strategy 3 (weighted summation method) in section 3.3 specifically in line 337-352. And we perform the experiments on the three strategies in section 4.2.

21) Section 4.2. presents the results, and the results are discussed in a different and much better way than the methods explained in the previous section was given. The authors performed several experiments using common datasets.  Even the English language is better. The authors only forgot to put the best results in bold (as promised in Captions of Tables 2, 5, 6, 7 and 8.

Answer: Thanks for your kind advice and comment for our manuscript. We have put the best results in bold in section 4.3.

22) Issue 3: Language is a problem, although it improves in Section 4.2. It is almost unreadable in Sections 1 – 4.1. It is not a major problem – their paper [A1] published in Entropy is not in perfect English either. Some examples: Authors tend to glue reference and abbreviation, making the text less readable: “retrieval(CBIR)[1-3]”, it should be  “retrieval (CBIR) [1-3]” (many examples). “we perform re-ranking[19]and query expansion(QE)[22] on …”. “outputted”. “Researchers tend to use the feature maps come from the last convolutional layer.” “NetVLAD which inspired by VLAD and produces (?∗?)∗? dimentional aggregated descriptors which can be directly used”

Answer: Thanks for your kind advice and comment for our manuscript. We have revised these errors in our manuscript and tried to promote readability of our manuscript. And we have tried our best to improve our language representation in English to enhance the readability of our manuscript.

23) In my opinion, the authors should quote the original Shannon paper, as it was a foundation of Information Theory as a scientific branch.

Answer: Thanks for your kind advice and comment for our manuscript. We have quoted the original Shannon paper in section 2.2 of line xx-xx. The reference number is [39].

We would like to express our great appreciation to you and reviewers for comments on our paper. Looking forward to hearing from you.

Thank you and best regards.

Yours sincerely,

Ping-ping Liu

Reviewer 2 Report

 In this paper, authors present a new pooling method, promoting the performance of R-MAC pooling. In some aspects the paper has a poor structure and needs a lot of improvements, more specifically:

Introduction section, besides the conventional theoretical background, should contain the research hypothesis which are consequently justified within the paper. Also, the last paragraph of the introduction section should contain at least brief overview of individual chapters of the paper.

Besides the used datasets, authors should consider testing the proposed method on medical images, having different parameters; therefore, effectivity of the proposed method may be different as well.

Authors should consider the image noise influence on the proposed method. Which type of the deterministic noise the most influence the effectivity. In this context, the image noise dynamic effect may be interesting to observe on the method’s effectivity.

Lastly, the conclusion section is too shortened and simplified. Authors should provide the conclusion in a critical way, highlighting all the pros and cons of the proposed method. In the ideal case, authors should add the discussion section, providing a critical view on the results.

Author Response

On behalf of my co-authors, we thank you very much for giving us an opportunity to revise our manuscript. We appreciate editor and reviewers very much for the positive and constructive comments and suggestions on our manuscript (ID: 611445).

We have studied reviewer’s comments carefully and have made revision. According to your advice, we amended the relevant part in manuscript. Some of your questions were answered as bellow.

1) Introduction section, besides the conventional theoretical background, should contain the research hypothesis which are consequently justified within the paper. Also, the last paragraph of the introduction section should contain at least brief overview of individual chapters of the paper.

Answer: Thanks for your kind advice and comment for our manuscript. We have revised the introduction section of our manuscript. We give the description of research hypothesis that the performance of image retrieval would be improved when taking the distribution information of feature maps into consideration to generate deep feature representations with more information.   And it is justified by the results of experiment. And we also represent the brief overview of individual chapters of our manuscript in line 95-98.

2) Besides the used datasets, authors should consider testing the proposed method on medical images, having different parameters; therefore, effectivity of the proposed method may be different as well.

Answer: We are very grateful to your kind comments for the manuscript. We have tested our proposed method on medical dataset. And it is composed by two public medical datasets of Brain_Tumor_Dataset and Origa in line 554-566. And we give the results in Table 10, which shows that our proposed FDE is effective in improving the performance of image retrieval. Furthermore, the results demonstrate that our proposed method of fusing FDE with R-MAC outperforms the existing art-of the-state methods.

3) Authors should consider the image noise influence on the proposed method. Which type of the deterministic noise the most influence the effectivity. In this context, the image noise dynamic effect may be interesting to observe on the method’s effectivity.

Answer: We are very grateful to your kind comments for the manuscript. We have discussed impact of different noises to our proposed FDE in line 311-322. We conduct a set of comparative experiments on computing the values of FDE for a set of 50 images distilled randomly from Oxford5k dataset and after applying four kinds of noises on these images. And we give the result in Figure 2.  We could conclude that the values of FDE calculated by our proposed method are hardly sensitive to poisson, salt and speckle noises. The guassian noise would like to increase the value of FDE slimly at few images. And it shows that our proposed FDE calculated by scheme 1 is robust to the most types of noise.

4) Lastly, the conclusion section is too shortened and simplified. Authors should provide the conclusion in a critical way, highlighting all the pros and cons of the proposed method. In the ideal case, authors should add the discussion section, providing a critical view on the results.

Answer: Thanks for your helpful comments. We have enriched the conclusion and given the pros and cons of method and further represented our following work plains. And we have added the discussion in section 4.4 and specifically in line 567-577.

We would like to express our great appreciation to you and reviewers for comments on our paper. Looking forward to hearing from you.

Thank you and best regards.

Yours sincerely,

Ping-ping Liu

Round 2

Reviewer 1 Report

The paper is considerably improved. The authors have replied to all the comments, and they have changed the paper accordingly.  I have an impression that the grammar could be improved, there are capital letters in the middle of the sentences (automatically inserted by the text processor, but not corrected).